# Biological Investigation of Amaryllidaceae Alkaloid Extracts from the Bulbs of *Pancratium trianthum* Collected in the Senegalese Flora

**DOI:** 10.3390/molecules26237382

**Published:** 2021-12-05

**Authors:** Seydou Ka, Natacha Mérindol, Insa Seck, Simon Ricard, Abdoulaye Diop, Cheikh Saad Bouh Boye, Karima Landelouci, Benoit Daoust, Lionel Berthoux, Geneviève Pépin, Matar Seck, Isabel Desgagné-Penix

**Affiliations:** 1Département de Chimie, Biochimie et Physique, Université du Québec à Trois-Rivières, Trois-Rivières, QC G8Z 4M3, Canada; Seydou.Ka@uqtr.ca (S.K.); Natacha.Merindol@uqtr.ca (N.M.); Simon.Ricard@uqtr.ca (S.R.); Benoit.Daoust@uqtr.ca (B.D.); 2Laboratoire de Chimie Organique et Thérapeutique, Faculté de Médecine, de Pharmacie, et d’Odontologie de Dakar, Dakar B.P. 5005, Senegal; insa1.seck@ucad.edu.sn (I.S.); matarsec@yahoo.fr (M.S.); 3Laboratoire Bactériologie-Virologie, CHU Aristide Le Dantec, Université Cheikh Anta Diop de Dakar, Dakar B.P. 5005, Senegal; laycoumba@yahoo.fr (A.D.); cheikh.boye@ucad.edu.sn (C.S.B.B.); 4Département de Biologie Médicale, Université du Québec à Trois-Rivières, Trois-Rivières, QC G8Z 4M3, Canada; Karima.Landelouci@uqtr.ca (K.L.); Lionel.Berthoux@uqtr.ca (L.B.); Genevieve.Pepin3@uqtr.ca (G.P.); 5Groupe de Recherche en Biologie Végétale, Université du Québec à Trois-Rivières, Trois-Rivières, QC G8Z 4M3, Canada

**Keywords:** alkaloid, Amaryllidaceae, *Pancratium trianthum*, antiviral, antimicrobial, dengue virus, HIV

## Abstract

Amaryllidaceae plants are rich in alkaloids with biological properties. *Pancratium trianthum* is an Amaryllidaceae species widely used in African folk medicine to treat several diseases such as central nervous system disorders, tumors, and microbial infections, and it is used to heal wounds. The current investigation explored the biological properties of alkaloid extracts from bulbs of *P. trianthum* collected in the Senegalese flora. Alkaloid extracts were analyzed and identified by chromatography and mass spectrometry. Alkaloid extracts from *P. trianthum* displayed pleiotropic biological properties. Cytotoxic activity of the extracts was determined on hepatocarcinoma Huh7 cells and on acute monocytic leukemia THP-1 cells, while agar diffusion and microdilution assays were used to evaluate antibacterial activity. Antiviral activity was measured by infection of extract-treated cells with dengue virus (DENV_GFP_) and human immunodeficiency virus-1 (HIV-1_GFP_) reporter vectors. Cytotoxicity and viral inhibition were the most striking of *P. trianthum*’s extract activities. Importantly, non-cytotoxic concentrations were highly effective in completely preventing DENV_GFP_ replication and in reducing pseudotyped HIV-1_GFP_ infection levels. Our results show that *P. trianthum* is a rich source of molecules for the potential discovery of new treatments against various diseases. Herein, we provide scientific evidence to rationalize the traditional uses of *P. trianthum* for wound treatment as an anti-dermatosis and antiseptic agent.

## 1. Introduction

The Amaryllidaceae family encompasses over 1600 species scattered all around the globe and is among the top 20 most considered medicinal plant families [1]. They are bulbous flowering plants also exploited for ornamental purposes. Approximately one-third of known Amaryllidaceae species grow in South Africa, and they are commonly used in folk medicine [2,3]. Traditional usage of Amaryllidaceae ranges from simple health problems (e.g., headache, cough, boils) to complicated diseases (e.g., cancer, tuberculosis, diabetes). Amaryllidaceae preparations are recognized for their antimicrobial, anti-tumoral, anti-acetylcholinesterase (AChE), and cytotoxic properties [4,5,6,7,8,9,10,11,12,13,14,15,16]. West African Amaryllidaceae species such as *Pancratium* sp., collected in Senegal for traditional medicine [17], have been scarcely studied [18]. The genus *Pancratium* contains approximately 20 species, extending from the Canary Islands through the Mediterranean region to tropical Asia, and from West Africa to Namibia [19]. *P. trianthum* Herb. (English name: pancratium lily; local names: baka, ngada, tondut) [20] plant extracts are traditionally used for irritation-calming, wound-healing, anti-oedema, anti-dermatosis, anti-septic, anti-epileptic, psychotropic, and fungicidal properties [17,21]. *P. trianthum* is considered to be toxic in Senegal and Sudan and thus is restricted to external usage only [5], while it is considered edible in Nigeria and Western Sahara [4,22]. 

Phytochemical studies have shown that the biological properties of Amaryllidaceae largely originate from a specific class of specialized metabolites that they produce, called the Amaryllidaceae alkaloids (AAs) [23,24,25]. AA classification describes nine different AA-types according to their structure and their biosynthesis pathway [24]. Galanthamine, a widely occurring AA, is currently used as an AChE inhibitor to treat Alzheimer‘s disease symptoms [26]. Several other pharmaceutically relevant AAs are under study, such as crinamine for the treatment of Parkinson’s disease [27] or cherylline for its antiviral properties [28,29]. Lycorine, crinine, and haemanthamine were previously described as the most abundant types of alkaloids in the *Pancratium* genus [30]. A study published in 1983 identified the AAs trisphaeridine, hippeastrine, hordenine, pancratine, tazettine, lycorine, galanthamine, and trianthine from aerial and underground plant parts of *P. trianthum* collected from the Botanical Garden of Pyatigorsk State Pharmaceutical Academy in Russia [31,32]. However, no study has reported on the AA content nor the biological activities of *P. trianthum* from native areas. Such study would help uncover its full therapeutic potential and possibly lead to the development of alternative pharmaceuticals to improve human health. Here, we continue our screening of biological activities of traditionally used native under-studied Amaryllidaceae from Senegal [28,29]. Thus, the current investigation explores the anti-AChE, cytotoxicity, antimicrobial, antiviral, and proinflammatory effects of alkaloid extracts from *P. trianthum* collected in Senegal.

## 2. Results

### 2.1. Species Phylogenetic Analysis

To confirm the species of the bulbs collected and used in this study, we amplified and sequenced DNA from a chloroplastic gene encoding the large chain of ribulose bisphosphate carboxylase *rbcL* [33]. The *P. trianthum rbcL* sequence obtained (Appendix A-Figure A1) was blasted in the National Center for Biotechnology Information (NCBI) database using BLASTn. Top hits were related to *rbcL* sequences from *Pancratium* species, with over 96% sequence identity. Phylogenetic analysis was carried out using phylogeny.fr [34], with default values, including *rbcL* sequences from 10 *Pancratium* species and 1 outgroup, from the species *Lilium lancifolium* Thunb., an Asparagales that belongs to the Liliaceae instead of the Amaryllidaceae family (Figure 1). Maximum likelihood analysis showed a monophyletic clade of the *rbcL* sequences from all *Pancratium* species, including *P. trianthum P. illyricum* L. *rbcL* sequences clustered outside this common node, and the outgroup *L. lancifolium* Thunb. was robustly separated from all Amaryllidacease species. *P. trianthum* sequences grouped closely with *P. canariense* Ker-Gawl, *P. zeylanicum* L., and *P. hirtum* A. Chev (Figure 1), consistently with a phylogenetic study on *Pancratium* species of the Mediterranean area, which included sequences from *P. trianthum* collected in Burkina Faso [33].

### 2.2. Phytochemical Analysis

Next, an acid–base extraction method was performed on bulbs of *P. trianthum* to isolate alkaloids from other organic compounds based on their acid–base properties. The extraction yielded 0.06% of the initial bulb biomass extracted. Alkaloid profiles were initially analyzed using chromatographic methods. TLC screening revealed at 254 nm, 365 nm using Dragendorff’s reagent showed the presence of different alkaloids with distinct R_f_ values of 0.6, 0.4, and 0.2 (Appendix A-Figure A2).

An optimized high-performance liquid chromatography (HPLC) with photodiode array (PDA) detector was performed to better characterize alkaloids. Three alkaloid standards (i.e., lycorine, galanthamine, and narciclasine) were used. Identification of the alkaloids extracted from bulbs of *P. trianthum* was accomplished by a comparison of the retention time (RT) and absorption spectrum with those obtained for the standards. The RT of lycorine, galanthamine, and narciclasine standards are respectively 7.19, 7.61, and 7.94 min in the standard solution (data not shown). HPLC analysis of the *P. trianthum* alkaloid extract showed peaks with RT corresponding to lycorine and narciclasine (Appendix A-Figure A2B), although for the latter the absorption spectrum was different compared with the narciclasine standard, suggesting that it was another alkaloid. No RT peak corresponding to galanthamine was detected in *P. trianthum* alkaloid extracts (Appendix A-Figure 2B). Interestingly, several unidentified peaks appeared with RTs of 6.46, 7.48, 7.92, 8.40, 8.86, and 15.40 min, suggesting potential alkaloids (Appendix A-Figure 2B).

*P. trianthum* alkaloid extracts were further investigated using GC-MS analysis, which led to the identification of six AAs by comparing the GC-MS spectra obtained with those available in NIST 05 database and in the literature (Table 1, Figure 2, Appendix A-Figure A3). Two structures could not be attributed to any previously reported alkaloid chemical profile. However, based on the specificity of the extraction method towards the isolation of alkaloids with respect to their physicochemical properties, the two unknown compounds might have been alkaloid. Furthermore, their RT and [M^+^] in the GC-MS analysis mimicked Amaryllidaceae alkaloids (Table 1).

Altogether, we observed six alkaloids classified according to their structure into lycorine (**1** and **2**), crinine (**3a/b**, **4**, and **5**), and narciclasine (**6**) ring types, together with two unidentified compounds (**7** and **8**) (Table 1). Compounds (**2**) and (**6**) showed ions at *m*/*z* 249, 248, 190, 163, 123, and 95 and at *m*/*z* 223, 193, 164, 138, and 111, respectively, which were not listed in NIST 05 database (Appendix A-Figure A3). However, identical fragmentations were reported in the literature and corresponded to 11,12-dehydroanhydrolycorine (**2**) and trisphaeridine (**6**) [13]. Hamayne (**5**), also not available in NIST 05 database, showed ions at *m*/*z* 287, 258, 242, 186, and 153 and was identified by similarity with the reported AA from *Rhodophiala pratensis* [36], while compound (**4**) displayed ions at *m*/*z* 273, 201, 175, 157, 141, and 128, identically to 8-*O*-demethylmaritidine from *Amaryllis belladonna* L. [35]. Additionally, an approximation of the relative proportion of identified AAs was estimated as a percentage of the total ion current (TIC) chromatogram. The major compounds (i.e., most abundant) were hamayne (**5**, 39% of TIC), 8-*O*-demethylmaritidine (**4**, 15% of TIC), and unidentified compound **8** (16% of TIC). Approximately 58% of identified alkaloids were crinine-type, 15% of lycorine-type, and 5% of the narciclasine-type (Table 1 and Figure 2).

### 2.3. In Vitro Anti-AChE Activity

*P. trianthum* plant preparations are used in folk medicine to treat central nervous system disorders. Therefore, we tested the anti-AChE (human) activity of our extracts. Galanthamine hydrobromide was used at a concentration of 3.7 µg/mL (10 µM) as a positive control, and extract dilutions matching DMSO concentrations were used as negative controls to normalize the effect of the solvent (Figure 3A). Galanthamine blocked 59% of acetylcholinesterase activity. *P. trianthum* alkaloid extracts inhibited the activity of AChE in a dose-dependent manner at concentrations ranging from 3.9 to 500 µg/mL, with an IC_50_ of 94 µg/mL (Figure 3A).

### 2.4. Cytotoxicity Activity

Several *Pancratium* species are cytotoxic plants with antiproliferative properties, and the AA lycorine detected in *P. trianthum* is notoriously cytotoxic [5]. The cytotoxic activity of *P. trianthum* alkaloid extracts was determined on THP-1 and Huh7 malignant cell lines, and the cytotoxic concentration of the extract causing a 50% reduction in cell viability (CC_50_) was calculated for each cell line (Figure 3B). Cell viability, as measured through ATP (adenosine triphosphate) levels, was strongly affected by the alkaloid extract in a dose-dependent manner at concentrations ranging from 0.02 to 2.5 µg/mL for both cell lines (Figure 3B). The alkaloid extracts were found to be cytotoxic down to 0.156 µg/mL in THP-1 and 0.313 µg/mL in Huh7, with CC_50_ values of 0.23 and 0.45 µg/mL, respectively (Figure 3B).

### 2.5. Antibacterial Activity

*P. trianthum* extracts are used in folk medicine as antiseptic to treat wounds, suggesting antimicrobial properties. Here, an agar diffusion assay was used to test the antibacterial activities of *P. trianthum* alkaloid extract. MICs were determined on different bacterial strains (Appendix A-Figure A4). The positive control cefotaxime was active at the tested concentration against *S. aureus* and *P. aeruginosa*, with 22 mm and 24 mm diameter growth inhibition areas, respectively. Cefotaxime MIC on *S. aureus* and *P. aeruginosa* was 2 µg/mL (data not shown). The highest antibacterial activity of the extracts was observed against *S. aureus*, with a 20 mm diameter growth inhibition area at the tested concentration of 16 mg/mL (Appendix A-Figure A4). Microdilution results confirmed that the alkaloid extract was active against Gram-positive (*S. aureus*) and against Gram-negative strains (*E. coli* and *P. aeruginosa*), though moderately, with MIC values ranging from 1 to 2 mg/mL (Appendix A-Figure A4), respectively, suggesting that *P. trianthum* holds weak antibacterial properties.

### 2.6. Antiviral Activity

Next, the antiviral activity of alkaloid extracts from *P. trianthum* was measured against lentivirus HIV-1 in THP-1 cells and dengue flavivirus in Huh7 cells (Figure 4). We used a pseudotyped HIV-1_GFP_ vector that infects cells, integrates into the cell genome, and expresses viral proteins and GFP. However, they did not produce nor release infectious viral particles. We treated THP-1 cells with 0.09 to 3.125 µg/mL of *P. trianthum* extract and infected them with HIV-1_GFP_ at an MOI = 1 (Figure 4A). GFP expression was measured 72 h post-infection. A dose-dependent inhibition of HIV-1 was observed with increasing concentrations of AAs extract. Weakly cytotoxic (Figure 3B) concentrations of 0.09 and 0.19 µg/mL significantly reduced pseudotyped HIV-1_GFP_ infectivity by 28% and 52%, respectively. Interestingly, an EC_50_ of 0.17 µg/mL was obtained (Figure 4A), a concentration at which viability was 65% that of the control (Figure 3B).

*P. trianthum* alkaloid extracts were then tested for their ability to inhibit dengue flavivirus DENV_GFP_ infection. The DENV_GFP_ vector is replication-competent, and thus able to propagate to neighboring cells, and produces GFP upon infection concomitant with translation of its genomic RNA. GFP expression was measured at 72 h post-infection (Figure 4B,C). Fluorescent infected cells were visualized on an inverted microscope, and their frequency was measured on a flow cytometer. Adenosine analog NITD008 was used as a positive control at 5 µM, whereas matching DMSO concentrations were used as negative controls for each dilution. DMSO treatment had no apparent effect on viral replication. Noteworthy, all tested concentrations ranging from 0.019 to 2.5 µg/mL significantly inhibited DENV_GFP_ infection in a dose-dependent manner (Figure 4B, 4C). No infected cells were detected upon treatment with alkaloid extracts at concentrations higher than 0.078 µg/mL, as monitored either by microscopy or flow cytometry. Flow cytometry analysis showed that the lowest tested extract concentration (0.019 µg/mL) yielded a 20% reduction in DENV infectivity compared with controls. It also confirmed that DENV replication was nearly completely blocked at 0.078 µg/mL, with only 3.3% of cells infected. These results demonstrate that alkaloids extracted from *P. trianthum* are active against dengue virus at non-cytotoxic concentrations, with an EC_50_ of 0.029 µg/mL (Figure 4B).

### 2.7. Pro-Inflammatory Activity 

The antiviral properties of *P. trianthum* extracts could be associated with a pro-inflammatory activity. Thus, IFN-type I induction by *P. trianthum* extract was investigated. IFN stimulation was measured in LL171 cell lines containing the interferon-stimulated response element (ISRE)–luciferase reporter. DMXAA, a STING (stimulator of interferon genes) activator, was used as positive control, whereas DMSO was used as negative control. When cells were treated with *P. trianthum* extract at concentrations ranging from 0.015 to 0.5 µg/mL, there was no detectable activation of luciferase expression, and hence no IFN production (Figure 5). However, when DMXAA was used in combination with *P. trianthum* extract, luciferase expression was triggered in a reverse dose-dependent manner. ISRE activity was increased by 2.7-fold compared with DMXAA treatment alone and by 14.4-fold compared with DMSO at the three lowest concentrations, (0.015, 0.031, and 0.0625 μg/mL), showing a potentiation between the extract and DMXAA. This suggests that low doses of *P. trianthum* extracts have a synergistic effect when used in combination with IFN-type I inducer DMXAA.

## 3. Discussion

In this study, alkaloid preparations extracted from *P. trianthum* bulbs collected in Senegal were investigated for their biological properties. TLC analysis of crude extracts showed different types of AAs, while HPLC analysis revealed lycorine and narciclasine-like AAs, along with additional peaks suggesting that other AAs were present in *P. trianthum* extracts. Alkaloid extracts were subjected to GC-MS analysis resulting in the identification of lycorine (**1**) and vittatine (**3a**)/crinine (**3b**) by comparison with their mass spectra available in the NIST05 database and of 11,12-dehydroanhydrolycorine (**2**), 8-*O*-demethylmaritidine (**4**), hamayne (**5**), and trisphaeridine (**6**) by comparing with the mass spectra available in the literature. As vittatine (**3a**) only differs from crinine (**3b**) in the position of the 5,10b-ethano bridge, which can only be distinguished by a circular dichroism spectrum, (**3**) could be either one or the other crinine-type of AAs. Using the percentage of TIC as approximative quantification, we observed that crinine- and lycorine-type AAs were the most abundant AAs in *P. trianthum* bulb extracts. Unidentified compounds represented approximately 16% of the crude extracts and should be isolated in future studies using additional chromatography techniques. In contradiction to a previous report [31], but consistent with the HPLC analysis, galanthamine, hippeastrine, hordenine, pancratine, tazettine, and trianthine were not detected in *P. trianthum* extracts. This difference could be explained by the fact that plants were not collected from the same environment as AA production and could be affected by environmental and climatic conditions. In addition, our study specifically targeted alkaloids extracted from bulb.

In Senegal, leaves and bulbs of *P. trianthum* are used to treat central nervous system disorder, heal wounds, and soothe irritations [17], suggesting potential anti-AChE, antibacterial, and antiviral activity. The AChE inhibitory activity is generally attributed to galanthamine and lycorine-types AAs [37,38]. Hamayne (**5**) detected in the extract was also previously reported to have weak AChE inhibitory activity [39]. In our study, *P. trianthum* alkaloid extracts weakly inhibited AChE activity in a dose-dependent manner, with concentrations ranging from 3.9 to 500 µg/mL. Although the observed levels of inhibition were lower than previously observed in *P. maritimum* L. [40,41], they were consistent with the lack of detection of galanthamine-type alkaloids in our extracts. 

Previous studies showed that lycorine (**1**), the main phenanthridine AA in our extract, displayed a strong antitumor activity [42,43]. *P. trianthum* alkaloid extracts were increasingly cytotoxic at concentrations above 0.313 µg/mL for hepatocarcinoma Huh7 cells and 0.156 µg/mL for monocytic leukemia THP-1 cells, consistent with the presence of the cytotoxic AA lycorine. *P. trianthum* bulb alkaloid extracts appeared to be more cytotoxic than reported from other *Pancratium* species such as *P. illyricum* L., but different cell lines were used, and the contents of their alkaloids are different [44]. 

Crinine- and lycorine-types have also been associated with antibacterial properties [45,46,47]. Hence, we tested the antibacterial potency of the extract. All tested strains were sensitive to high concentrations of the alkaloid extract. The Gram-positive species *S. aureus* was more strongly inhibited than Gram-negative species (*E. coli* and *P. aeruginosa*). This activity could be specific to limited *Pancratium* species, such as *P. illyricum* L. alkaloid extracts that were described to be completely ineffective against both Gram-positive and Gram-negative bacteria [42]. Thus, detected vittatine (**3a**)/crinine (**3b**) and lycorine (**1**) might contribute to the antimicrobial effect of the alkaloid extract. 

While the rate of new HIV-1 infection is declining and the coverage of people receiving antiretroviral therapy (ART) is growing in Senegal, prevalence remains high in specific populations such as sex workers (4.8% infected, of which 28.3% are under ART) and men who have sex with men (27.6%, of which 37.8% are under ART) [48]. Because of the high prevalence and the constant threat of the virus escaping treatment due to mutation-associated resistance, there is still a strong impetus to identify new antiviral compounds. In some studies, lycorine (**1**) has been shown to inhibit HIV-1 infection [49,50]. Our results show that low concentrations of *P. trianthum* extracts reduce HIV-1 infectivity levels, although full inhibition was only observed at cytotoxic concentrations. To continue our screening of *P. trianthum* antiviral potential, we then measured its effect on DENV infection. Several flaviviruses cocirculate in tropical and subtropical areas of the world and threaten the lives of hundreds of millions of people, making the development of broad-spectrum anti-flaviviral compounds a necessity. Previous studies showed that some Amaryllidaceae extracts present anti-dengue potential [28,51], and the AA lycorine (**1**) detected in this study potently inhibits flaviviruses in in vitro and in vivo models [52,53,54]. Here, we show that non-cytotoxic concentration of *P. trianthum* extracts ranging from 0.019 to 0.156 µg/mL displayed strong antiviral activity, with full inhibition at 0.156 µg/mL. When compared with our recent study on *Crinum jagus* (J. Thomps.) Dandy extract, the *P. trianthum* antiviral activity was 8.6-fold higher (IC_50_ = 0.25 μg/mL vs. 0.029 μg/mL, respectively) [28]. Interestingly, low doses of *P. trianthum* extracts also displayed potentiation effects with IFN-inducers. Thus, purification of alkaloids from *P. trianthum* could possibly lead to the discovery of strong antiviral compounds.

## 4. Materials and Methods

### 4.1. Plant Materials, Chemicals, and Species Identification

*P. trianthum* bulbs were collected in Saint Louis, Senegal (16°3′19.35″ N, 16°25′42.25″ W), in December 2018. Collected plants were taxonomically identified using the database at the Herbarium of IFAN at the Université Cheikh Anta Diop of Dakar. 

In addition, genomic DNA was extracted from dried roots of *P. trianthum* using the DNeasy plant mini kit (QIAGEN) according to the manufacturer’s instructions. Yield and purity of total extracted DNA were quantified with a Nanodrop (IMPLEN, QC, CA) and stored at −20 °C for later use. Ribulose-bisphosphate carboxylase gene (*rbcL*) DNA barcode of *P. trianthum* was amplified by PCR using TaKaRa’s PrimeSTAR GXL Premix kit with primers (F-5′-GGATTACCAGCCTTGATCG-3′ and R-5′-TTCACGAGCAAGATCACGTC-3′) [33]. The PCR mixture (20 µL) contained 2 µL of genomic DNA, 10 µL Takara mix, 0.4 µL of each primer (10 µM, forward and reverse primers), and 7.2 µL of ultrapure-DNase free water. The thermocycler program consisted of an initial denaturation step (98 °C for 2 min), followed by 30 amplification cycles (98 °C for 10 s, 55 °C for 15 s, and 68 °C for 75 s). After PCR, 5 μL of amplified product was loaded on a 1% agarose gel, and specific size amplicons (1044 bp) were sequenced using both forward and reverse primers.

### 4.2. Crude Alkaloids Extraction and TLC Analysis

Alkaloid extraction was achieved using the method described in [55]. *P. trianthum* dried bulbs (50 g) were crushed manually and macerated for 24 h with MeOH at room temperature, and the macerate was filtered and concentrated under reduced pressure. This crude extract was acidified with sulphuric acid (2%) at pH = 2 and extracted successively with Et_2_O (4 × 200 mL) and EtOAc (4 × 200 mL) to remove neutral material. The resulting acidic aqueous solution was basified with concentrated ammonia (25%) up to pH = 10, then extracted with EtOAc (4 × 200 mL) and evaporated. Alkaloid extracts were then dissolved in EtOAc at a final concentration of 1 mg/mL and used for thin-layer chromatography (TLC). Qualitative analysis of alkaloids was performed on TLC silica gel 60 F_254_ aluminum sheets 20 × 20 cm, (Merck, Darmstadt, Germany). The TLC plate was eluted with MeOH: EtOAc (25:75 *v*/*v*), dried at room temperature, observed under UV light at 254 nm and 365 nm, and then revealed with Dragendorff reagent [56,57].

### 4.3. HPLC-PDA and GC-MS Analysis of Alkaloid Extracts

For the HPLC analysis, we followed the method described by [58], with some modifications. The alkaloid extract was dissolved in MeOH at a final concentration 0.5 mg/mL. Afterward, 10 µL of each sample and 10 µL of 100 ppm of each standard (galanthamine, narciclasine (both from Tocris Bioscience, Bristol, UK), and lycorine (Sigma-Aldrich, St. Louis, MO, USA)) were injected and analyzed on a Shimadzu Prominence-I LC-2030C with diode array detector (PDA). For separation, the Kinetex C18 column (150 × 4.6 mm^2^, 5 µm particle size; Phenomenex) was used. HPLC oven temperature was set at 40 °C. A gradient elution with two solvents, 1% ammonium acetate buffer (solvent A) and 100% acetonitrile (solvent B), was used. The 90:10 B to A solvent gradient ratio was first maintained for 10 min, then shifted to 69:31 over 1 min, 70:30 over 4 min, and finally 90:10 over 3 min. After 18 min, A was increased to 90%, and B was reduced to 10% for 5 min, yielding a total run time of 23 min.

For the GC-MS analysis, 1 mg/mL of alkaloid extract in MeOH was directly injected into the GC-MS (Agilent Technologies 6890N GC coupled with 5973N inert MSD) in EI (Electron Ionization) mode at 70 eV. The temperature ramp used is described as follows: temperature was set at 100 °C for 2 min, followed by 100–180 °C at 15 °C min^−1^, 180–300 °C at 5 °C min^−1^, and a 10 min hold at 300 °C. Injector and detector temperatures were set at 250 °C and 280 °C, respectively, and the flow rate of carrier gas (He) was 1 mL min^−1^. A split ratio of 1:10 was applied, and the injection volume was 1 µL [59]. Alkaloids were identified by comparison with the 2005 National Institute of Standards database based on matching mass spectra, GC-MS spectra of authentic compounds previously isolated and identified by other spectroscopic methods in these species, or with data obtained from the literature. The total ion current (TIC) percentage provided in Table 1 was connotated with the proportion of each compound in the extract as a semi-quantitative estimate. The area of the GC-MS peaks depends both on the concentration of the related compounds and their relative signal intensity in MS.

### 4.4. In Vitro Acetylcholinesterase (AChE) Inhibition Assay

*In vitro* inhibition of electric eel, *Electrophorus electricus*, AChE by the alkaloid extract of *P. trianthum* was assessed using the method described in [29]. 

### 4.5. Cell Lines

Human hepatocarcinoma cell line Huh7 (kindly provided by Professor Hugo Soudeyns, CHU Sainte-Justine, Montréal, QC, Canada) and murine LL171 reporter cells [60] were maintained in Dulbecco’s modified Eagle’s medium high glucose (DMEM), supplemented with 10% fetal bovine serum (FBS) and 1% penicillin–streptomycin (PS, all from Cytiva Hyclone) solution. Human acute monocytic leukemia cell line THP-1 was grown in Roswell Park Memorial Institute Medium (RPMI) containing 10% FBS and 1% PS. All cells were maintained at 37 °C and 5% CO_2_. 

### 4.6. Cytotoxicity Assay

Cell viability was evaluated using the Cell-Titer GLO assay kit (Promega, Madison, WI, USA). Briefly, 50 µL of 7.5 × 10^3^ Huh7 cells/well or 2 × 10^4^ THP-1 cells/well was seeded in 96-well dark plates and cultured for 16 h. Then, they were treated with 50 µL of alkaloid extract at concentrations (<0.5% DMSO) ranging from 0.019 to 2.5 µg/mL for 72 h. Afterward, 100 µL of room temperature Cell-Titer GLO reagent was added in each well to room temperature-equilibrated plates. Plates were rocked for 2 min, and the luminescence signal was measured 10 min later using a microplate spectrophotometer (Synergy H1, Biotek, QC, Canada). The percentage of viable cells was calculated at each concentration. All assays were performed in triplicate.

### 4.7. Bacteria and Viruses

Three bacteria species including the Gram-positive strain *Staphylococcus aureus* ATCC 29213 and two Gram-negative strains, *Escherichia coli* ATCC 35218 and *Pseudomonas aeruginosa* ATCC 27853, were obtained from the American Type Culture Collection (ATCC) and from the Laboratory of Bacteriology-Virology at Aristide Le Dantec Hospital (Senegal). Bacteria were cultured in Mueller Hinton (MH) agar media and incubated at 37 °C for 24 h before use.

Dengue virus vectors expressing green fluorescent protein (DENV_GFP_ [61]) and single-round infection pseudotyped human immunodeficiency virus-1 (HIV-1_GFP_ [62]) were used to investigate antiviral activity. The multiplicity of infection (MOI) of HIV-1_GFP_ was assessed by measuring the infectivity of serially diluted vector preparation in Crandell-Rees Feline Kidney (CRFK) cells, while DENV_GFP_ titer was measured by plaque assay as described in [28].

### 4.8. Agar Diffusion Assay

Antimicrobial activity of the alkaloid extract was first studied using the agar diffusion assay method. A suspension of each strain was prepared into sterile physiological water to obtain a final inoculum, estimated at 1.5 × 10^8^ CFU/mL according to 0.5 McFarland turbidity. Alkaloid extracts were dissolved in DMSO at 16 mg/mL. For the assay, a sterile cotton swab was immersed in the inoculum, then wrung on the wall of the tube. The swab was then spread over on the agar plate to obtain uniform inoculum. Wells were made on Mueller Hinton agar plates using a sterile cylinder of 6 mm diameter. Plates were dried for 5 min, and 100 μL of alkaloid extract was deposited. Plates were incubated at 37 °C for 18–24 h. The antibacterial activity of the alkaloid extract was then measured as an inhibition zone surrounding the well [63]. Cefotaxime was used as a positive control.

### 4.9. Broth Dilution Method for Determination of Minimal Inhibitory Concentration (MIC)

Microdilution of alkaloid extracts from *P. trianthum* was performed using a modification of Balouiri et al. [64]. Dilutions were started by pipetting 100 μL of alkaloid extract into the first well of a 96-well plate containing 100 μL of MH broth. Serial dilutions were then carried out to obtain a range of concentrations between 0.03 to 8 mg/mL. Then, 10 μL of bacterial suspension cultures was added into each well. Plates were incubated at 37 °C for 24 h. The MIC (minimal inhibitory concentration) was determined as the lowest concentration of the alkaloid extract that completely suppressed the growth of microorganisms (which was determined by the wells showing no turbidity). Tested bacteria were exposed to broth without the alkaloid extract as a control.

### 4.10. In Vitro DENV_GFP_ Infectivity Assay

Briefly, Huh7 cells were seeded at 1.5 × 10^4^ cells per well in 48-well plates, cultured for 16 h, and then pretreated with concentrations of *P. trianthum* alkaloid extract ranging from 0.02 to 2.5 µg/mL, infected with DENV_GFP_ 2 h later at a multiplicity of infection (MOI) of 0.1 PFU/cell and incubated at 37 °C for 72 h. Green fluorescence signal of infected cells was visualized and pictured on a Axio Observer microscope (Carl Zeiss, Inc., Toronto, ON, Canada). Cells were trypsinized and fixed in 4% aqueous formaldehyde, then processed for flow cytometry analysis of GFP expression using a FC500 MPL cytometer (Beckman Coulter, Inc., CA) coupled with the FCS Express 6 software (De Novo Software, Pasadena, CA, USA). The adenosine analogue NITD008 (10 µM) was used as a positive control for inhibition of DENV infection. Extract-matched concentrations of DMSO were used as a negative control. All assays were performed in triplicate at least twice.

### 4.11. In Vitro Pseudotyped HIV-1_GFP_ Infectivity Assay

The antiretroviral activity of *P. trianthum* Herb.’s crude extract was evaluated using pseudotyped HIV-1_GFP_ in THP-1 cells. THP-1 cells were seeded at 2.0 × 10^4^ cells per well in 96-well plates and incubated overnight. Cells were treated with 5 concentrations of extract (from 0.1 to 1.56 µg/mL) and then infected with HIV_GFP_ at a MOI at 1. After 72 h, the percentage of infected cells was measured using a FC500 MPL cytometer (Beckman Coulter, Inc., Brea, CA, USA) and analyzed using FlowJo software (FlowJo LLC, Ashland, OR, USA). Matched concentrations of DMSO were used as a negative control. All assays were performed in triplicate.

### 4.12. In Vitro Proinflammatory Assay

In vitro proinflammatory assay was performed using the Luciferase Assay Systems kit (Promega). Briefly, 200 µL LL171 cells was seeded at 1.5 × 10^4^ cells/well in 96-well plates and cultured for 16 h. Then, the medium was replaced with medium containing alkaloid extract at a concentration ranging from 0.015 to 0.5 µg/mL for 24 h. The supernatant was removed, and cells were rinsed with PBS. Then, 30 µL of lysis buffer was added and homogenized. Afterward, 20 µL of each lysate was transferred into 96-well opaque plates, 100 µL of Luciferase Assay Reagent (Promega) was added, and luminescence was measured at 480 nm using a microplate spectrophotometer (Synergy H1, Biotek, Quebec, Canada). 5,6-Dimethylxanthenone-4-acetic acid (DMXAA, 20 µg/mL) was used as a positive control. All assays were performed in triplicate.

## 5. Conclusions

In conclusion, this study led to the detection of eight AAs, six of which were identified by GC-MS, in the alkaloid extract of *P. trianthum* Herb.’s bulb collected in Senegalese flora. Based on the traditional use of the *Pancratium* genus for wound-healing, central nervous system disorder, and antiproliferative and antiviral purposes, the alkaloid extracts prepared were screened for antibacterial, anti-AChE, cytotoxic, and antiviral properties. Alkaloid extracts displayed antibacterial effect, with MIC values of 1 mg/mL for *S. aureus* and 2 mg/mL for *E. coli* and *P. aeruginosa* but weak anti-AChE property (IC_50_ = 94 µg/mL). Interestingly, the *P. trianthum* extracts displayed strong and moderate antiviral activity against DENV_GFP_ and pseudotyped HIV-1_GFP_, with EC_50_ of 0.029 and 0.17 µg/mL, respectively. We conclude that the medicinal properties of *P. trianthum* may be attributed to its alkaloid components and provide the scientific basis for its traditional use to prevent infections. Finally, this study supports the role of Amaryllidaceae species as a source of compounds with potential therapeutical applications.

## Figures and Tables

**Figure 1 molecules-26-07382-f001:**
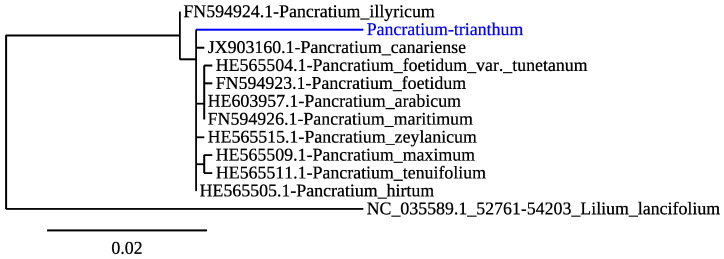
Maximum likelihood phylogenetic analysis of ribulose-bisphosphate carboxylase gene (*rbcL)* DNA sequences from *P. trianthum* (blue) collected from Senegal. Reference sequences were identified using the GenBank accession numbers. The *rbcL* sequence from *Lilium lancifolum*, an Asparagales belonging to Liliaceae but not Amaryllidaceae family, was used as outgroup. The scale of branch length = 0.02 (2% of genetic variation is shown in the bottom of the tree).

**Figure 2 molecules-26-07382-f002:**
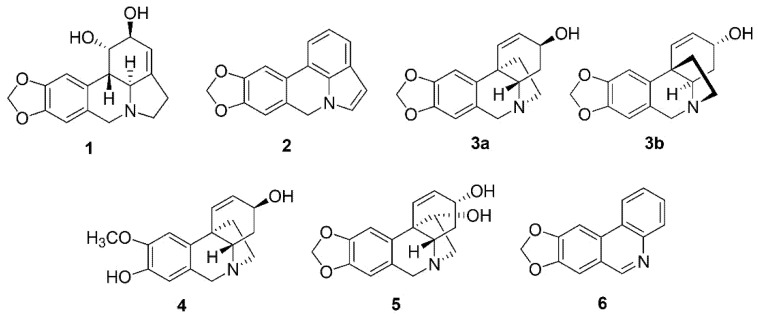
Structures of identified AAs confirmed by GC-MS analysis. (**1**) Lycorine; (**2**) 11,12-Dehydroanhydrolycorine; (**3**) Vittatine/Crinine; (**4**) 8-*O*-Demethylmaritidine; (**5**) Hamayne; (**6**) Trisphaeridine.

**Figure 3 molecules-26-07382-f003:**
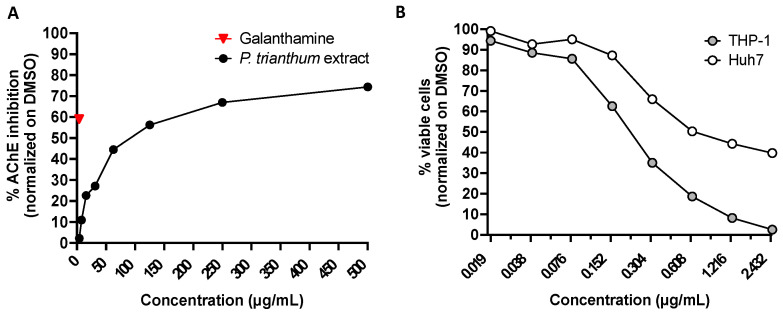
Anti-acetylcholinesterase and cytotoxic activity of AA bulb extracts from *Pancratium trianthum* (**A**) Anti-acetylcholinesterase (human)activity. The percentage of inhibition of acetylcholinesterase activity was calculated using DMSO as a negative control. Galanthamine hydrobromide (3.7 µg/mL or 10 μM) was used as a positive control. (**B**) Cellular ATP levels were measured in Huh7 and THP-1 cells to assess viability following 72 h of incubation.

**Figure 4 molecules-26-07382-f004:**
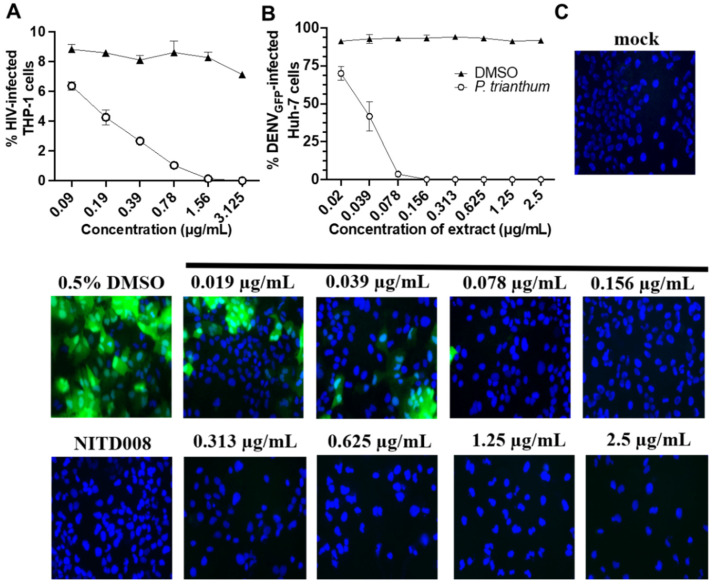
Antiviral activity of *P. trianthum* alkaloid extracts. (**A**) Inhibition of HIV-1 infection. THP-1 cells were treated with increasing concentrations of alkaloid extracts and then infected with VSV-G pseudotyped HIV-1_GFP_. (**B**) Inhibition of DENV_GFP_ replication. Huh7 cells treated with *P. trianthum* alkaloid extracts were infected with DENV_GFP_ and measured by flow cytometry. For (**A**,**B**), infection levels were measured at 72 h post-infection and shown are means of triplicates with standard deviation; the x axis is in log2 scale. (**C**) Inhibition of DENV_GFP_ infection in Huh7 cells treated with *P. trianthum* alkaloid extracts as observed by inverted fluorescence microscopy after 72 h of infection. Representative images are shown with cell nuclei stained with Hoechst 33342 (blue) and DENV-infected cells (green). DMSO (vehicle) was used as a negative control.

**Figure 5 molecules-26-07382-f005:**
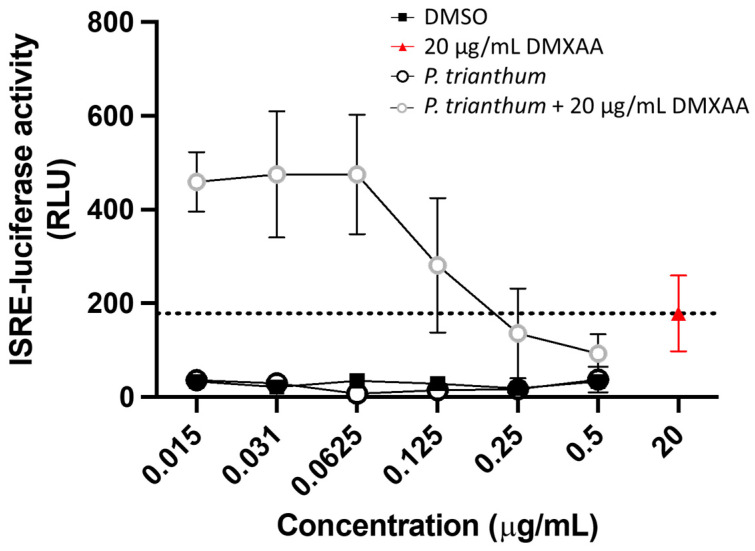
Pro-inflammatory properties of *P. trianthum* alkaloid extracts. Type I IFN activation was measured through the expression of luciferase driven by the ISRE (IFN-stimulated response element) promoter element and detected by luminescence. LL71 cells were treated with 20 µg/mL DMXAA (a STING activator; red triangle) and *P. trianthum* bulb extracts at concentrations ranging from 0.015 to 0.5 µg/mL, separately (black circles) or in combination with 20 µg/mL DMXAA (gray circles). Matched concentrations of DMSO (black squares) were used as a negative control, respectively. Means of triplicates with standard deviation of luminescence measured 24 h after treatment are shown.

**Table 1 molecules-26-07382-t001:** Alkaloids identified by GC-MS in *P. trianthum* bulb extracts. Values are expressed as percentages of total ion current (TIC).

Ring Type	Alkaloid	[M^+^]	B.P.	R.T. (min)	TIC (%)	Identification Source
Lycorine	Lycorine (**1**)	287	226	25.902	5.9	NIST 05 Database
	11,12-Dehydroanhydrolycorine (**2**)	249	248	23.811	9.1	[13]
Crinine	Vittatine/Crinine (**3a/3b**)	271	271	21.886	3.8	NIST 05 Database
	8-*O*-Demethylmaritidine (**4**)	273	273	22.275	15.4	[35]
	Hamayne (**5**)	287	258	25.321	39.1	[36]
Narciclasine	Trisphaeridine (**6**)	223	223	19.022	5.0	[13]
unknown	Unidentified (**7**)	287	223	23.547	5.2	n/a
unknown	Unidentified (**8**)	279	278	27.682	16.4	n/a

BP, base peak; RT, retention time (in minute); TIC, total ion current; n/a, not applicable.

## Data Availability

Not applicable.

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
