# Peer review of "Biological Investigation of Amaryllidaceae Alkaloid Extracts from the Bulbs of Pancratium trianthum Collected in the Senegalese Flora"

_molecules, 2021, doi:10.3390/molecules26237382_

Round 1
Reviewer 1 Report
The results obtained give credence to the traditional uses of Pancreatium trianthum in Senegal. Although most of the tested biological activities were weak, the anti-viral results look promising. The data presented are a useful addition to knowledge and worth recording.
However, consideration should be given to the following:
- I find it strange that there is not one reference to the work of Cahlikova and colleagues. This Czech group has contributed greatly to the knowledge of alkaloid distribution, separation, characterization, and biological activity of Amaryllidaceae alkaloids.
- It states in the Introduction that the test species has been investigated previously (references 23, 24), but the alkaloids reported in these studies are not mentioned. This should be rectified as a comparison of these results with those obtained in the present study is of relevance .
- The botanical authority of the test species should be included when the species name is first used, but after that it should be omitted.
- In some places US spelling is used and in others English. Consistency is required.
- In the cytotoxicity study, two tumor cell lines were used, but not one normal, non-tumor cell line. This is something that the authors should perhaps take into account in subsequent studies.
- In my opinion, Figures 2A and 2B are not needed and could be removed. These depict very routine work.
- Similarly, Figures 4A and 4B are not required as such work is routine and all the relevant results are included in the text.

Author Response
Point by point Response to reviewer’s :
DETAILED RESPONSES TO REVIEWER’S COMMENTS
(The responses are provided in blue italicized and bold font type)
Reviewer #1 - Review Report Form
Open Review
(x) I would not like to sign my review report
( ) I would like to sign my review report
English language and style
( ) Extensive editing of English language and style required
(x) Moderate English changes required
( ) English language and style are fine/minor spell check required
( ) I don't feel qualified to judge about the English language and style
|
Yes |
Can be improved |
Must be improved |
Not applicable |
|
|
Does the introduction provide sufficient background and include all relevant references? |
( ) |
(x) |
( ) |
( ) |
|
Is the research design appropriate? |
(x) |
( ) |
( ) |
( ) |
|
Are the methods adequately described? |
(x) |
( ) |
( ) |
( ) |
|
Are the results clearly presented? |
(x) |
( ) |
( ) |
( ) |
|
Are the conclusions supported by the results? |
(x) |
( ) |
( ) |
( ) |
Comments and Suggestions for Authors
The results obtained give credence to the traditional uses of Pancreatium trianthum in Senegal. Although most of the tested biological activities were weak, the anti-viral results look promising. The data presented are a useful addition to knowledge and worth recording.
However, consideration should be given to the following:
- I find it strange that there is not one reference to the work of Cahlikova and colleagues. This Czech group has contributed greatly to the knowledge of alkaloid distribution, separation, characterization, and biological activity of Amaryllidaceae alkaloids.
RESPONSE: We wish to thank the reviewer for the comments and suggestions to improve our manuscript which are truly appreciated. Several references were added including some from Cahlikova group.
- It states in the Introduction that the test species has been investigated previously (references 23, 24), but the alkaloids reported in these studies are not mentioned. This should be rectified as a comparison of these results with those obtained in the present study is of relevance.
RESPONSE: We thank the reviewer for this comment. The isolated alkaloids are added in the introduction. The discussion was also modified : In contradiction to a previous report , galanthamine, hippeastrine, hordenine, pancratine, tazettine, and trianthine were not detected in P. trianthum extracts. This difference could be explained by the fact that plants were not collected from the same environment (Russia botanical garden vs wild growth in Senegal) and AA production could be affected by environmental and climatic conditions In addition, our study targeted specifically alkaloids extracted from bulb.
- The botanical authority of the test species should be included when the species name is first used, but after that it should be omitted.
RESPONSE: It has been corrected.
- In some places US spelling is used and in others English. Consistency is required.
RESPONSE: English Canadian spelling is now used throughout the text for consistency.
- In the cytotoxicity study, two tumor cell lines were used, but not one normal, non-tumor cell line. This is something that the authors should perhaps take into account in subsequent studies.
RESPONSE: We thank the reviewer for this suggestion that we will take into account for our future studies.
- In my opinion, Figures 2A and 2B are not needed and could be removed. These depict very routine work.
RESPONSE: Figure 2A and 2B were moved to supplementary data.
- Similarly, Figures 4A and 4B are not required as such work is routine and all the relevant results are included in the text.
RESPONSE: Figure 4A and 4B were moved to supplementary data.
Reviewer 2 Report
There are a lot of studies about AA and therefore this article is not anything original. I like a lot of biological testes, but it is always better to get pure substances and do a number of tests on them than to test only the extract. With the extract, it is difficult to find out what substance is behind the effect therefore they are only suitable for screening.
In my opinion, the identification of the substances in this article is not sufficient.
- why weren't standards already used in TLC and GC/MS?
- some alkaloids cannot be determined without further testing, such as: vittatine (It is vittatine or crinine? It depends on the optical rotation.
- How do you know that unidentified substances are alkaloids?
- How much crude and alkaloid extracts were obtained? Would it be possible to isolate some substances?
Biological tests:
I completely did not understand the ACHE measurement method.
Why do you use galanthamine in concentration 10 µM as positive control? I think it would be more appropriate to use galanthamine at its IC50 concentration as a standard for efficacy comparisons. I would also recommend reporting all concentrations in the same units, ie in µg/ml. So you have a badly marked galanthamine brand in the graph, because 10 µM is not 10 µg/ml. I am also missing information about the material used. Do you use pure alkaloid galanthamine or hydrobromide? Do you use human AChE from blood or recombinant, or from eel? In my opinion, inhibition 94 µg/ml is not significant.
The cytotoxic assay lacks a standard.
Figure 4 A is not meaningless without scale. I would not include it in the article, moreover, when the inhibition was very low.
I recommend correcting gross errors and better evaluating the results with respect to the measured values and comparisons with standards or results of other research groups.
Author Response
Point by point Response to reviewer’s :
DETAILED RESPONSES TO REVIEWER’S COMMENTS
(The responses are provided in blue italicized and bold font type)
Reviewer #2 - Review Report Form
Open Review
(x) I would not like to sign my review report
( ) I would like to sign my review report
English language and style
( ) Extensive editing of English language and style required
( ) Moderate English changes required
( ) English language and style are fine/minor spell check required
(x) I don't feel qualified to judge about the English language and style
|
Yes |
Can be improved |
Must be improved |
Not applicable |
|
|
Does the introduction provide sufficient background and include all relevant references? |
(x) |
( ) |
( ) |
( ) |
|
Is the research design appropriate? |
(x) |
( ) |
( ) |
( ) |
|
Are the methods adequately described? |
( ) |
(x) |
( ) |
( ) |
|
Are the results clearly presented? |
( ) |
( ) |
(x) |
( ) |
|
Are the conclusions supported by the results? |
( ) |
( ) |
(x) |
( ) |
Comments and Suggestions for Authors
There are a lot of studies about AA and therefore this article is not anything original. I like a lot of biological testes, but it is always better to get pure substances and do a number of tests on them than to test only the extract. With the extract, it is difficult to find out what substance is behind the effect therefore they are only suitable for screening.
RESPONSE: We wish to thank the reviewer for the comments and suggestions to improve our manuscript which are truly appreciated. We agreed with the reviewer’S comment about it is best to test pure compound. Amaryllidaceae alkaloids biological activities are indeed intensively studied. West-African medicinal plants, and in particular the literature on the native Pancratium trianthum alkaloids and their biological activities was lacking. Moreover, only one study from our group have described one Amaryllidaceae species (Crinum jagus) possessing anti-flaviviral potential, and in this case, the activity was impressively high. We agree with the reviewer that further studies to purify the alkaloids will be important. However, in our opinion, we must start with plant extracts compounds profiles and their biological activities first and publish (such as 10.1371/journal.pone.0213049, 10.1038/s41598-020-64026-z, 10.1186/s13065-021-00781-y, 10.3390/molecules26113452, 10.3390/molecules26113227 and 10.3390/molecules26051318). Then, the purification of single molecules can be achieved, if the potential bioactivity is interesting for future application such as to help improve human health.
In my opinion, the identification of the substances in this article is not sufficient.
- why weren't standards already used in TLC and GC/MS?
RESPONSE: We did not use standard for routine qualitative TLC analysis because the few pure standard we have are rare and expensive. We keep them for HPLC and MS analyses.
Also for the TLC experiment, we wanted to see the orange color revealed with the Dragendorff reagent, as alkaloids precipitate in the acidic medium in the presence of Dragendorff to yield the orange color. To ensure that the orange spots observed in the TLC analyses are alkaloids, we performed an HPLC analysis using standards such as galanthamine, lycorine and narciclasine. In order to identify more peaks observed in HPLC that are potentially alkaloids, we used GC-MS. With this method, we were able to identify alkaloids described in the literature and those available in the NIST 05 library. To address the reviewer’s concern about these results, figures 2A and 2B were removed from the main figures of the manuscript and placed in the supplementary data section.
- some alkaloids cannot be determined without further testing, such as: vittatine (It is vittatine or crinine? It depends on the optical rotation.
RESPONSE: We thank the reviewer for this comment. We agree with the reviewer, vittatine and crinine only differ in the position of the 5,10b-ethano bridge, and they can only be distinguished by a circular dichroism spectrum. To answer this, we added crinine to the legend of Figure 2 and in the table for more accuracy, as well of the following sentence in the discussion: As vittatine only differs from crinine in the position of the 5,10b-ethano bridge which can only be distinguished by a circular dichroism spectrum, they could not be differentiated in this study.
- How do you know that unidentified substances are alkaloids?
RESPONSE: We do not know for sure and corrected the manuscript to clarify this point.
We hypothesized that since we used a specific acid-base extraction method (often used for alkaloids and Amaryllidaceae alkaloids), we most likely obtained alkaloids based on their physicochemical properties. In addition, the unknown compounds displayed an RT and [M+] in the GC-MS analysis that mimic Amaryllidaceae alkaloids.
- How much crude and alkaloid extracts were obtained? Would it be possible to isolate some substances?
RESPONSE: The extraction yield was low: 0.06% of the initial mass. Unfortunately, at the moment, this project is stopped. We agree with the reviewer that it would be interesting to collect more bulbs of Pancratium trianthum from Senegal and to isolate molecules with potent antiviral properties in future studies.
Biological tests:
I completely did not understand the ACHE measurement method.
Why do you use galanthamine in concentration 10 µM as positive control? I think it would be more appropriate to use galanthamine at its IC50 concentration as a standard for efficacy comparisons. I would also recommend reporting all concentrations in the same units, ie in µg/ml. So you have a badly marked galanthamine brand in the graph, because 10 µM is not 10 µg/ml. I am also missing information about the material used. Do you use pure alkaloid galanthamine or hydrobromide? Do you use human AChE from blood or recombinant, or from eel? In my opinion, inhibition 94 µg/ml is not significant.
RESPONSE: We thank the reviewer for this comment. A single concentration 3.7 ug/ml of galanthamine hydrobromide was used as a positive control. In our previous study we determined IC50 of galanthamine hydrobromide was approximatively 2.8 ug/mL (doi: 10.1016/j.phytochem.2020.112390). The information was added to the text. The figure 3A was corrected to answer the reviewer’s concern. We used a modified protocol of the Acetylcholinesterase Assay Kit from Abcam that contains human AChE, as better referred to in the method section. We agree that the activity is weak and clarified this point in the discussion.
The cytotoxic assay lacks a standard.
RESPONSE: The objective of this experiment was to evaluate cell viability to determine the concentrations to be tested for antiviral experiments. Thus, DMSO, the solvent, was used as a negative control.
Figure 4 A is not meaningless without scale. I would not include it in the article, moreover, when the inhibition was very low.
RESPONSE: Figure 4A was moved to supplementary data.
I recommend correcting gross errors and better evaluating the results with respect to the measured values and comparisons with standards or results of other research groups.
RESPONSE: We revised the entire manuscript. As there are no reports on the biological effects of Pancratium trianthum, we included citations of reports on other species biological assays. Moreover, our study is the first one on anti-dengue and anti-HIV activity for such species.

Round 2
Reviewer 2 Report
Thank you for incorporating vittatin / crinin, but the formula does not show which one it is. It is not possible to mark it this way. List both variants. Some formulas are not painted correctly. In the case of a dashed constraint, all associated constraints must be below the plane.
Figure 2 would be better moved behind GC / MS analysis.
Figure 3 - graph A is still wrong. Put the red mark for galanthamine in the correct place and indicate the units in µg / mL
more about comments of figure in the word document
In the section Crude alkaloids extraction, and TLC analysis, it would be useful to write how many g / mg of extract was obtained even though it is only 0.06 %. (To improve the yield, I recommend extraction by boiling in methanol or ethanol. If you do not want to use high temperature, use ultrasound for 30 minutes.)
There are no articles on Pancratium trianthum, but on all the substances you have identified, yes.
You have corrected the AChE inhibition claim in the discussion, but in the end you still report strong inhibitory activity. I do not agree with the statement that 8 AA was detected in the study (in conclusion). Nowhere in the text did I find an explanation for the unknown substances as to why they should be alkaloids, as stated in RESPONSE.
I recommend reading the whole article again and correcting all errors, shortcomings and inconsistencies, not only those that I mention here.
All errors and inconsistencies reduce the level of this article. If everything is carefully corrected and adjusted to the correct results, this article can be better evaluated. So far, it can be seen that the author did not put much work into the correction (based on comments).
more about comments of figure in the word document

Author Response
Point by point Response to reviewer’s :
DETAILED RESPONSES TO REVIEWER’S COMMENTS
(The responses are provided in blue italicized and bold font type)
Reviewer #2 - Review Report Form - Round 2
Open Review
(x) I would not like to sign my review report
( ) I would like to sign my review report
English language and style
( ) Extensive editing of English language and style required
( ) Moderate English changes required
( ) English language and style are fine/minor spell check required
(x) I don't feel qualified to judge about the English language and style
|
Yes |
Can be improved |
Must be improved |
Not applicable |
|
|
Does the introduction provide sufficient background and include all relevant references? |
(x) |
( ) |
( ) |
( ) |
|
Is the research design appropriate? |
(x) |
( ) |
( ) |
( ) |
|
Are the methods adequately described? |
(x) |
( ) |
( ) |
( ) |
|
Are the results clearly presented? |
( ) |
( ) |
(x) |
( ) |
|
Are the conclusions supported by the results? |
( ) |
( ) |
(x) |
( ) |
Comments and Suggestions for Authors
Thank you for incorporating vittatin / crinin, but the formula does not show which one it is. It is not possible to mark it this way. List both variants. Some formulas are not painted correctly. In the case of a dashed constraint, all associated constraints must be below the plane.
RESPONSE: We wish to thank the reviewer for the comments and suggestions to further improve our manuscript which are truly appreciated. The reviewer is right about the necessity of these corrections. We added the crinine structure and corrected the others according to the reviewers suggestions.
Figure 2 would be better moved behind GC / MS analysis.
RESPONSE: We moved figure 2 after Table 1.
Figure 3 - graph A is still wrong. Put the red mark for galanthamine in the correct place and indicate the units in µg / mL more about comments of figure in the word document
RESPONSE: We apologize to the reviewer for not having included the corrected figure at the first review. It was corrected in the revised manuscript.
In the section Crude alkaloids extraction, and TLC analysis, it would be useful to write how many g / mg of extract was obtained even though it is only 0.06 %. (To improve the yield, I recommend extraction by boiling in methanol or ethanol. If you do not want to use high temperature, use ultrasound for 30 minutes.)
RESPONSE: We are grateful to the reviewer for this suggestion that we will test during our next extraction process. We included the yield in the phytochemical section.
There are no articles on Pancratium trianthum, but on all the substances you have identified, yes.
RESPONSE: The reviewer is correct. The originality of our findings relies on the detection of these alkaloids in P. trianthum and on the strong antiviral activity of its extract.
You have corrected the AChE inhibition claim in the discussion, but in the end you still report strong inhibitory activity. I do not agree with the statement that 8 AA was detected in the study (in conclusion). Nowhere in the text did I find an explanation for the unknown substances as to why they should be alkaloids, as stated in RESPONSE.
RESPONSE: We have replaced significant by weak line 483: In our study, P. trianthum alkaloid extracts weakly inhibited AChE activity in a dose dependent manner with concentrations ranging from 3.9 to 500 µg/mL.
I recommend reading the whole article again and correcting all errors, shortcomings and inconsistencies, not only those that I mention here.
RESPONSE: We went through the entire manuscript and improved our discussion.
All errors and inconsistencies reduce the level of this article. If everything is carefully corrected and adjusted to the correct results, this article can be better evaluated. So far, it can be seen that the author did not put much work into the correction (based on comments).
more about comments of figure in the word document
RESPONSE: We thank the reviewer for his time and his suggestions to better our article.
